# A Novel Method to Assess Antimicrobial Susceptibility in Commensal Oropharyngeal *Neisseria*—A Pilot Study

**DOI:** 10.3390/antibiotics11010100

**Published:** 2022-01-13

**Authors:** Jolein Gyonne Elise Laumen, Saïd Abdellati, Christophe Van Dijck, Delphine Martiny, Irith De Baetselier, Sheeba Santhini Manoharan-Basil, Dorien Van den Bossche, Chris Kenyon

**Affiliations:** 1STI Unit, Department of Clinical Sciences, Institute of Tropical Medicine, 2000 Antwerp, Belgium; cvandijck@itg.be (C.V.D.); sbasil@itg.be (S.S.M.-B.); ckenyon@itg.be (C.K.); 2Laboratory of Medical Microbiology, University of Antwerp, 2610 Wilrijk, Belgium; 3Clinical and Reference Laboratory, Department of Clinical Sciences, Institute of Tropical Medicine, 2000 Antwerp, Belgium; sabdellati@itg.be (S.A.); idebaetselier@itg.be (I.D.B.); dvandenbossche@itg.be (D.V.d.B.); 4Department of Microbiology, Laboratoire Hospitalier Universitaire de Bruxelles-Universitair Laboratorium Brussel (LHUB-ULB), Université Libre de Bruxelles, 1000 Brussels, Belgium; delphine.martiny@lhub-ulb.be; 5Faculté de Médecine et Pharmacie, Université de Mons, 7000 Mons, Belgium; 6Division of Infectious Diseases and HIV Medicine, University of Cape Town, Cape Town 7700, South Africa

**Keywords:** *Neisseria* species, antimicrobial susceptibility, surveillance

## Abstract

Commensal *Neisseria* provide a reservoir of resistance genes that can be transferred to the pathogens *Neisseria gonorrhoeae* and *N. meningitidis* in the human oropharynx. Surveillance programs are thus needed to monitor resistance in oropharyngeal commensal *Neisseria,* but currently the isolation and antimicrobial susceptibility testing of these commensals is laborious, complex and expensive. In addition, the posterior oropharyngeal/tonsillar swab, which is commonly used to sample oropharyngeal *Neisseria*, is poorly tolerated by many individuals. We evaluated an alternative non-invasive method to isolate oropharyngeal commensal *Neisseria* and to detect decreased susceptibility to azithromycin using selective media (LBVT.SNR) with and without azithromycin (2 µg/mL). In this pilot study, we compared paired posterior oropharyngeal/tonsillar swabs and oral rinse-and-gargle samples from 10 participants and demonstrated that a similar *Neisseria* species diversity and number of colonies were isolated from both sample types. Moreover, the proportion of *Neisseria* colonies that had a decreased susceptibility to azithromycin was similar in the rinse samples compared to the swabs. This pilot study has produced encouraging data that a simple protocol of oral rinse-and-gargle and culture on plates selective for commensal *Neisseria* with and without a target antimicrobial can be used as a surveillance tool to monitor antimicrobial susceptibility in commensal oropharyngeal *Neisseria.* Larger studies are required to validate these findings.

## 1. Introduction

Commensal *Neisseria* are an important part of the normal oropharyngeal human microbiome [1]. Various lines of evidence suggest that the prevalence of antimicrobial resistance in commensal *Neisseria* is increasing over time [2,3,4,5,6,7,8,9]. This finding is worrisome, as commensal *Neisseria* provide a reservoir of resistance genes that can be horizontally transferred to pathogenic bacteria, such as *Neisseria gonorrhoeae* and *N. meningitidis,* via transformation [9,10,11,12,13]. As an example, it has recently been shown that the acquisition of sections of the gene encoding the mtrCDE efflux pump by *N. gonorrhoeae* from commensal *Neisseria* played a crucial role in the emergence of macrolide resistance in *N. gonorrhoeae* [14]. These findings have led to calls to establish surveillance programs that monitor antimicrobial susceptibility in commensal *Neisseria* [2,10,15].

Such a surveillance program faces a number of challenges. Currently, surveillance of antimicrobial susceptibility of commensal *Neisseria* involves taking oropharyngeal swabs and culturing the bacteria on non-selective plates. Most studies use Matrix-Assisted Laser Desorption/Ionization-Time-of-Flight mass spectrometry (MALDI-TOF MS) for further bacterial identification, and antimicrobial susceptibility methods, such as agar dilution or gradient dilution, need to be carried out to determine minimal inhibitory concentrations (MICs) [2,4,8].

This procedure is laborious, complex and expensive and would not be feasible to implement in places with limited access to specialized equipment, such as MALDI-TOF MS in low-income settings. In this pilot study, we therefore assessed two innovations in study methodology that would simplify surveillance of decreased susceptibility of non-pathogenic *Neisseria* in the human oropharynx: (i) using an oral rinse instead of a swab to sample the bacteria and (ii) using *Neisseria*-selective agar with and without an antimicrobial.

(i)There are a number of problems with using a posterior oropharyngeal/tonsillar swab to sample oropharyngeal *Neisseria*. First, this technique is unlikely to collect commensal *Neisseria* in other niches in the oral cavity, such as the tongue and buccal mucosa, which have been shown to be commonly colonized by commensal *Neisseria* [16]. Secondly, a posterior oropharyngeal/tonsillar swab is poorly tolerated. In one study, this sampling technique provoked an unpleasant gag reflex in more than half of the individuals [17,18]. In this study, the authors suggested that the absence of a gag reflex in the other individuals was caused by the inability to sample the posterior pharynx. As a result, not inducing a gag response was independently associated with a lower probability of detecting *N. gonorrhoeae* [17]. Additionally, a large variation in the detection rate of *N. gonorrhoeae* was found between the clinicians taking the samples, indicating a large inter-individual variability in accuracy [17]. A recent publication found that oral rinse samples are well tolerated and provide excellent results in terms of characterizing oral *Neisseria* species using whole genome sequencing data [19]. Our first innovation thus involved testing if oral rinse samples were able to detect the presence of commensal *Neisseria* as well as the currently recommended posterior oropharyngeal/tonsillar swab.(ii)As isolation, identification and assessment of antimicrobial susceptibility of commensal *Neisseria* on non-selective agar plates is laborious, we used LBVT.SNR agar plates selective for commensal *Neisseria* to reduce the workload and need for expensive equipment, such as MALDI-TOF-MS machines [20]. Additionally, instead of assessing the antimicrobial susceptibility of each individual colony, we assessed the proportion of commensal *Neisseria* that could grow on LBVT.SNR medium with 2 µg/mL azithromycin. This approach has been successfully used for a number of bacterial species to estimate the proportion of bacteria with antimicrobial resistance in a sample [21,22,23].

In this pilot study, we obtained paired posterior oropharyngeal/tonsillar swabs and oral rinse-and-gargle samples from 10 individuals and compared *Neisseria* species diversity and the number of colonies isolated between both sample types. Additionally, we compared the numbers of commensal colonies growing on a *Neisseria*-selective medium plate with and without azithromycin to establish the proportion of oropharyngeal *Neisseria* commensals with azithromycin resistance.

## 2. Materials and Methods

### 2.1. Study Design

This pilot study involved 10 employees of the Institute of Tropical Medicine in Antwerp, Belgium. The participants were recruited in July 2021. Candidates who used an antimicrobial over the last 6 months were excluded.

### 2.2. Sample Collection

All participants provided written informed consent prior to data and sample collection. Two different oropharyngeal samples were collected. A study physician rubbed both tonsillar pillars and the posterior oropharynx with an ESwab^TM^ and then placed the swab in the Eswab^TM^ medium (COPAN Diagnostics Inc., Brescia, Italy). We refer to this specimen as the swab sample. In addition, the participants were instructed to rinse/gargle their mouths with 15 mL sterile phosphate buffered saline (PBS, Oxoid^TM^, Dulbecco A) for 60 s, after which they were collected in a sterile container. Specifically, under direct observation, they rinsed their mouth for 15 s, then gargled for 15 s before repeating the rinse and gargle once more. This sample is termed the rinse sample. The samples were processed within 5 h.

### 2.3. Sample Processing

Ten-fold dilutions were made from both samples; 100 µL of the 10^−1^, 10^−2^ and 10^−3^ dilutions in PBS were inoculated on commensal *Neisseria* selective medium (LBVT.SNR: prepared according to the instructions below) with and without azithromycin (2 µg/mL, SigmaAldrich, Steinheim am Albuch, Germany). Plates were incubated for up to 48 h at 37 °C in 5–7% carbon dioxide. Isolates were defined as resistant to azithromycin if growth was detected on the plates with azithromycin.

The concentration of 2 µg/mL was chosen as azithromycin resistance in *N. gonorrhoeae* is defined according to the epidemiological cut-off value (ECOFF) of MIC > 1 mg/L, and the median azithromycin MIC of commensal *Neisseria* in a previous study with employees of the Institute of Tropical Medicine in Antwerp was 3.0 mg/L (IQR 2.0–4.0) [8].

For the preparation of LBVT.SNR plates, 1% Bacto-Tryptone (Difco, BD), 0.5% yeast extract (SigmaAldrich, Steinheim am Albuch, Germany), 0.5% sodium chloride, and 1.5% Bacto-Agar (Difco, BD) were dissolved. Neutral Red (0.3% wt/vol, SigmaAldrich, SigmaAldrich, Steinheim am Albuch, Germany) at a concentration of 5 mL/L was added and the medium was sterilized by autoclaving for 15 min at 121 °C [20]. After cooling to 56 °C, sucrose was added to the base medium to a final concentration of 1%, and vancomycin and trimethoprim were each added to a final concentration of 3.0 µg/mL [20].

### 2.4. Clinical Isolate Organism Identification

For every plate, one colony from each morphologically different phenotype was purified for further identification. Isolates were identified to the species level using MALDI-TOF MS, on a MALDI Biotyper^®^ Sirius IVD system using the MBT Compass IVD software and library (Bruker Daltonics, Bremen, Germany) as described previously [8]. Isolates identified as *N. macacae* were grouped into one category with *N. mucosa,* whereas isolates identified as *N. perflava* and *N. flavescens* were grouped into one category with *N. subflava* [24].

### 2.5. Data Analysis

The number of different *Neisseria* species was calculated as the number of all *Neisseria* species detected on each plate, per sample type. The number of colonies was determined by counting the number of colonies on the plate without azithromycin containing 20–200 colonies and normalized for the inoculated sample volume. The proportion of azithromycin-resistant *Neisseria* commensals was determined by dividing the number of colonies of the azithromycin-containing plate by the number of colonies detected on the plate without azithromycin, with a maximum of 100%.

Differences in the number of *Neisseria* species and colonies identified and the proportion with azithromycin resistance between the rinse and swab samples were analyzed using the Wilcoxon matched-pairs signed-rank test. These analyses were performed using STATA V13.

### 2.6. Ethical Considerations

The study was conducted according to the guidelines of the Declaration of Helsinki and approved by the Institutional Review Board of the Institute of Tropical Medicine (protocol code: 1351/20, date of approval: 21 April 2021). The Informed Consent Form (ICF) documents were designed in accordance with the requirements of the Helsinki Declaration (2013), the E6 ICH GCP Guidelines (2016) and the Belgian Law on Experiment on the Human Person (2004). All participants provided written informed consent before any study-related procedures were performed.

## 3. Results

### 3.1. Participants

The median age of the 10 participants was 45 years (IQR 45–55) and 60% were female.

### 3.2. Number of Different Neisseria Species

In total, 5 different *Neisseria* species were isolated: *Neisseria bacilliformis*, *Neisseria elongata*, *Neisseria oralis*, *Neisseria macacae*/*Neisseria mucosa* and *Neisseria subflava*/*Neisseria flavescens*/*Neisseria perflava*. There was no significant difference in the number of species identified via oral rinse (median 2 (IQR 2–2)) versus swabbing (median 2 (IQR 1–2), *p* = 0.250). The most notable difference between the two methods was that in three participants (2, 7, 10) an additional *Neisseria* species was isolated in the oral rinse that was not detected in the swab samples (Table 1, Appendix A).

### 3.3. Colony Count

There was no significant difference in the number of colonies isolated from the oral rinse samples (median 2.9 × 10^5^ (IQR 1.1 × 10^5^–8.7 × 10^5^) CFU/mL) versus swab samples (median 1.8 × 10^5^ (IQR 1.3 × 10^4^–5.0 × 10^5^) CFU/mL, *p* = 0.131) on the LBVT-SNR plates without azithromycin (Table 2, Appendix A). In two participants, less than 20 *Neisseria* colonies were cultured from the swab sample, while significant growth was obtained from the oral rinse sample.

### 3.4. Azithromycin Resistance in Commensal Neisseria

The proportion of colonies with azithromycin resistance did not differ significantly between the sample types (rinse sample: median 64.2% (IQR 46.3–100); swab sample: median 41.2% (IQR 20.3–70.0); *p* = 0.131; Table 2). In two participants (2 and 8), the proportion of azithromycin-resistant commensal *Neisseria* in the swab sample could not be determined, as there was no growth on either plate.

*Neisseria oralis* and *Neisseria elongata* were only detected on the plates without azithromycin, suggesting their susceptibility to this agent. On the other hand, *N. mucosa and N. subflava* were repeatedly isolated from azithromycin-containing (2 mg/L) plates. In one participant, *N. bacilliformis* was isolated from an oral rinse sample on the azithromycin-containing plate, but not on the plate without azithromycin.

### 3.5. Detection of Bacterial Species Other Than Neisseria

None of the pathogenic *Neisseria, N. gonorrhoeae* or *N. meningitidis,* were detected. In one participant, *Moraxella* was detected from the oral rinse sample, while in another participant, *Brevundimonas* species were detected in the swab and the oral rinse sample.

## 4. Discussion

In this pilot study, we compared *Neisseria* diversity and the number of colonies isolated from a swab sample and an oral rinse sample from 10 individuals. We were able to isolate a similar number of *Neisseria* species from the less invasive oral rinse compared to the swab sample. The *Neisseria* species isolated in this study were similar to those found in other studies using the conventional methodology of swab/culture/MALDI-TOF MS/MIC determination. In this study, *N. subflava/N. flavescens/N. perflava, N. macacae/N. mucosa, N. elongata, N. bacilliformis* and *N. oralis* were isolated in 100%, 60%, 20%, 10% and 10% of the participants, respectively. Previously, the commensal *Neisseria* species *N. subflava* (96.9%), *N. mucosa* (25.0%), *N. oralis* (25.0%), *N. cinerea* (9.4%), *N. elongata* (9.4%), *N. lactamica* (6.3%) and *N. bacilliformis* (3.1%) were isolated from the general Belgian population [8]. In a study isolating *Neisseria* species from pharyngeal specimens from 207 men who have sex with men in Vietnam, 5 different *Neisseria* species were isolated. In descending order of prevalence, these were: *N. subflava/N. flavescens/N. perflava* (80%)*, N. macacae/N. mucosa* (7.2%), *N. cinerea* (2.6%), *N. oralis* (1.5%) and *N. lactamica* (0.4%) [2]. The fact that we did not detect species such as *N. cinerea* and *N. lactamica* in our current study could be due to the low prevalence of these species and the small sample size in our study.

We found that the proportion of *Neisseria* colonies that were resistant to azithromycin was similar in the two different sample types. In most participants, the proportion of resistant colonies was higher in the oral rinse sample compared to the rinse sample. These differences might be explained by the fact that an oral rinse samples a larger surface of the mouth and pharynx than a swab that only samples the tonsils and posterior pharynx. As a result, the rinse sample may contain additional *Neisseria* species from different ecological niches than those isolated from the swab sample.

Only a limited number of studies have characterized the azithromycin susceptibility of oropharyngeal commensal *Neisseria.* Data from previous studies in Belgium suggest an increase in azithromycin MICs over time, high MICs for azithromycin in the general Belgian population and an even higher MIC distribution in a cohort of STI clinic clients with high antimicrobial consumption (median MIC 7.0 (IQR 3.0–280.2)) [4,8]. In these studies, antimicrobial susceptibility of commensal *Neisseria* was determined by plating ESwabs^TM^ on blood agar and identifying *Neisseria*-compatible colonies by Gram staining and oxidase tests [4]. Subsequently, species identity was confirmed by MALDI-TOF MS and E-tests were performed to determine MICs. This is a laborious and time-consuming process that requires access to a MALDI-TOF MS device. Furthermore, picking a single colony of each morphologically different type does not give a complete picture of the susceptibility range of a specific species, as MICs can vary considerably within one species. Using a simplified protocol, we were able to determine the proportion of commensal *Neisseria* colonies with an azithromycin MIC > 2 mg/L.

An important limitation of this study is the small sample size, which means it is underpowered to detect smaller effect sizes. Another limitation of this study was that we isolated and purified a single colony per morphologically different group for further species identification. This may have led to us missing particular *Neisseria* (and other) species that have a similar morphology to the sampled colonies. This limitation, however, also applies to the conventional method of profiling commensal susceptibilities. Using this approach, we only identified two non-*Neisseria* species: *Moraxella* and *Brevundimonas*, each in one participant, indicating that LBVT-SNR medium is an acceptable agar for the selection of commensal *Neisseria*.

This pilot study has produced encouraging data that a simple protocol of oral rinse and culture on LBVT.SNR plates with and without a target antimicrobial can be used as surveillance tool to monitor antimicrobial susceptibility in commensal *Neisseria*. Larger studies are required to validate these findings.

## Figures and Tables

**Table 1 antibiotics-11-00100-t001:** Different *Neisseria* species isolated (*n*) from paired oral rinse and oropharyngeal swab samples present on the azithromycin-containing plate (2 µg/mL) and the plate without azithromycin.

Participant	Oral Rinse Sample	Swab Sample
1	n = 2; *Neisseria subflava* *Neisseria macacae*	n = 2; *Neisseria subflava* *Neisseria macacae*
2	n = 3; *Neisseria subflava* *Neisseria macacae* ***Neisseria bacilliformis***	n = 2; *Neisseria subflava* *Neisseria macacae*
3	n = 2; *Neisseria subflava* *Neisseria macacae*	n = 2; *Neisseria subflava* *Neisseria macacae*
4	n = 2; *Neisseria subflava* *Neisseria macacae*	n = 2; *Neisseria subflava* *Neisseria macacae*
5	n = 2; *Neisseria subflava* *Neisseria elongata*	n = 2; *Neisseria subflava* *Neisseria elongata*
6	n = 2; *Neisseria subflava* *Neisseria macacae*	n = 2; *Neisseria subflava* *Neisseria macacae*
7	n = 3; *Neisseria subflava* *Neisseria macacae* ***Neisseria elongata***	n = 2; *Neisseria subflava* *Neisseria macacae*
8	n = 1; *Neisseria subflava*	n = 1; *Neisseria subflava*
9	n = 1; *Neisseria subflava*	n = 1; *Neisseria subflava*
10	n = 2; *Neisseria subflava* ***Neisseria oralis***	n = 1; *Neisseria subflava*

Species isolated in only one of the two sampling techniques are indicated in bold font.

**Table 2 antibiotics-11-00100-t002:** Number of colonies isolated from oral rinse and swab samples on the LBVT-SNR plates without azithromycin (expressed as CFU/mL) and the proportion of colonies with an azithromycin MIC higher than 2 µg/mL (%). Behind each sample type, the dilution that was used for colony counting is shown (resulting in 20–200 colonies after inoculation).

Participant	Oral Rinse CFU/mL (%)	Dilution Counted	Swab CFU/mL (%)	Dilution Counted
1	10^5^ (46)	10^−2^	10^4^ (70)	10^−1^
2	10^4^ (7)	10^−1^, 10^−2^	0	/
3	10^6^ (50)	10^−3^	10^4^ (20)	10^−1^
4	10^5^ (79)	10^−3^	10^5^ (100)	10^−3^
5	10^4^ (100)	10^−2^	10^6^ (23)	10^−3^
6	10^6^ (100)	10^−3^	10^6^ (23)	10^−3^
7	10^5^ (60)	10^−3^	10^5^ (70)	10^−3^
8	10^5^ (35)	10^−2^	0	/
9	10^5^ (69)	10^−2^	10^4^ (59)	10^−2^
10	10^5^ (100)	10^−3^	10^5^ (76)	10^−3^

/ is noted when less than 20 colonies were obtained after inoculating the dilution 10^−1^.

## Data Availability

All deidentified data are available as Appendix A to this manuscript. Additional related documents such as the study protocol, laboratory analysis plan, and informed consent form can be obtained from the corresponding author upon reasonable request.

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
