# Peer review of "A Novel Method to Assess Antimicrobial Susceptibility in Commensal Oropharyngeal Neisseria—A Pilot Study"

_antibiotics, 2022, doi:10.3390/antibiotics11010100_

Round 1

Reviewer 1 Report

Main objection.

Very small number of participants and strains obtained. With such a number of strains obtained, can we compare the methods of isolating bacteria of the genus Neisseria? With such a small number of participants and the obtained strains, it is difficult to relate to the effectiveness of the method, and thus try to look for statistical correlations.

Minor objection.

Sample processing

When describing the substrate used, please specify precisely what the concentration of the indicator was and what it was dissolved in.
Selected elements remain in many parts of the manuscript. Please format the text.
Information about the consent of the relevant committee that approved the study, as well as information about consent from patients should be in the Participants section.

Reviewer 2 Report

line 98 - please provide the manufacturer

line 105 - please provide the manufacturer

lines 144-150 - please use italics for the names of the microorganisms

Table 1 - why were only 7 participants included? please complete or explain 

Table 1 - under the table, please indicate the meaning of the bold

Reviewer 3 Report

  In the article “ A novel method to assess antimicrobial susceptibility in commensal oropharyngeal Neisseria – a pilot study” the authors presented  an alternative non-invasive method to isolate oropharyngeal commensal Neisseria, and to detect decreased susceptibility to azithromycin using selective media (LBVT.SNR) with and without azithromycin.

Two important limitations of this study are  the small sample size and  that it was purified a single colony per morphologically different group for further species identification.

As a recommendation , a high number of samples must be evaluated and from at risk patients.

Round 2

Reviewer 1 Report

Thank you for taking into account and responding to my comments.

I have no more comments.

Reviewer 3 Report

The authors modified the article in according with my recommendation .